# Implementing routine collection of EQ-5D-5L in a breast cancer outpatient clinic

Sofia Torres[1]*, Ahmed M. Bayoumi[1,2], Ana B. K. Abrahao[3], Maureen Trudeau[4,5], Kathleen I. Pritchard[5], Chun Nim Li[5], Nicholas Mitsakakis[6,7], Geoffrey Liu[8], Murray Krahn[1,9†]

1 Institute of Health Policy, Management and Evaluation, University of Toronto, Toronto, Ontario, Canada, 2 Li Ka Shing Knowledge Institute, MAP Centre for Urban Health Solutions, St. Michael's Hospital, Toronto, Ontario, Canada, 3 Hospital Samaritano Higienopolis–Oncologia Américas, São Paulo, Brazil, 4 Department of Medical Oncology, Odette Cancer Center, Sunnybrook Health Sciences Centre, Toronto, Ontario, Canada, 5 Sunnybrook Research Institute, Toronto, Ontario, Canada, 6 Children's Hospital of Eastern Ontario Research Institute, Ottawa, Ontario, Canada, 7 Division of Biostatistics, Dalla Lana School of Public Health, University of Toronto, Toronto, Ontario, Canada, 8 Department of Medicine, Princess Margaret Cancer Centre/University Health Network and University of Toronto, Division of Medical Oncology and Hematology, Toronto, Ontario, Canada, 9 Toronto Health Economics and Technology Assessment Collaborative, University Health Network, Toronto, Ontario, Canada

† Deceased.
* sofia.cidtorres@mail.utoronto.ca

**Data Availability Statement:** Data cannot be shared publicly because in accordance with the study's consent form, participant information collected will be handled in a confidential manner

## Abstract

### Purpose

A cross-sectional study was conducted to investigate the feasibility of implementing routine collection of the Euro-Qol 5 dimensions (EQ-5D) questionnaire, to inform drug and health technology reimbursement decision making.

### Methods

Women with breast cancer were recruited during scheduled clinic visits to an academic cancer centre. EQ-5D-5L was self-administered using electronic tablets. Diagnostic and treatment data were abstracted from patient charts. Feasibility was assessed primarily by the proportion of patients who fully completed EQ-5D-5L and by their willingness to complete the instrument at each clinic visit.

### Results

588 women were approached for study participation, 341 were enrolled. Fully completed EQ-5D-5L questionnaires were obtained in 323 participants (95% of participants, 95% CI 92–97%). Median time for EQ-5D-5L completion was 1.5 minutes (range:0.35 to 14.7). Mean age of participants was 58 years old. Most women who completed EQ-5D were White, born outside Canada and presented a high education level; one-quarter had metastatic disease. Most participants reported "No problems" in all EQ-5D-5L dimensions. Mean EQ-5D-5L index and mean EQ-5D-5L VAS values for all participants were respectively 0.83 (SD 0.13) and 75.7 (SD 17.45), with patients with metastatic disease scoring the lowest values. Seventy-eight percent of participants were willing to complete EQ-5D-5L at each clinic

and will be stored on site in a secure, privacy protected data warehouse, developed under the support of The Canada Foundation for Innovation at Sunnybrook Research Institute. As such, study data should not be transferred to a public repository. Data are available for researchers who meet the criteria for access to confidential data. For data access please contact the Sunnybrook Research Ethics Board at 416-480- 6100 x 688144 or Reb@sunnybrook.ca.

**Funding:** The authors received no specific funding for this work.

**Competing interests:** The authors have declared that no competing interests exist.

visit; lower Charlson comorbidity index and higher education level were predictors of willingness to continue to answer EQ-5D-5L.

## Conclusions

Tablet-based collection of EQ-5D-5L in the context of routine clinical practice proved to be feasible. However, many patients declined study participation or reported being in full health, raising concerns about whether this method of collecting EQ-5D adequately represents the health status of all breast cancer patients.

## Introduction

Breast cancer is the most frequent cancer in the world among women [1]. In developed countries, survival has improved significantly in the last decades, as a result of screening programs and more effective treatments [2–5]. Traditional outcome measures, such as overall survival, are central to cancer decision-making. However, as the number of breast cancer survivors increases, up-to-date real-world health-related quality of life (HRQOL) data can provide additional information regarding the overall burden of cancer and the effectiveness of medical interventions [6–8].

Preference-based HRQOL measures, in the form of utilities, are used by many countries in economic evaluations that inform drug and health technology reimbursement decision making [9–12]. Utilities are used to calculate survival, adjusted for quality of life, as Quality-Adjusted Life Years (QALYs) [13–16]. Cost per QALY estimates are often sensitive to utility values. However, utility studies often have small sample sizes and use a range of methods [17]. A sustainable and methodologically rigorous approach to collecting preference-based HRQOL measures at a population-level, across the full spectrum of a disease, could have broad benefits for epidemiological and economic analyses.

In Ontario, the Canadian province with the greatest population (13.5 million), regional cancer centres have successfully implemented routine standardized electronic symptom and functional status screening as part of routine care for ambulatory patients, demonstrating that patient-reported outcome (PRO) collection at population-level, with the primary purpose of improving individual patient care, is feasible [18].

In this study, we investigated the feasibility of administering the EuroQol 5-dimensions (EQ-5D) questionnaire, a generic measure of health status, in the context of routine clinical practice, in a cancer centre in Ontario. In this context, the purpose of EQ-5D collection was to obtain data to inform drug and health technology reimbursement decision making. EQ-5D scores, although available to clinicians, were not intended to be used for improving individual patient care. Other studies have investigated EQ-5D collection, but have significant differences in terms of objectives, methodology, population included and setting, when compared to our study. They either included the general population or patients with other diseases, used trained research assistants to help administer the questionnaires, discussed EQ-5D results with patients, assessed both patients´ and doctors' perspectives or the equivalence of paper and electronic versions of EQ-5D, among other differences [19–21].

Thus, we were interested in assessing feasibility in the context when data collection was not anticipated to have direct individual benefit for respondents. Feasibility was judged by examining several outcomes, including completeness of EQ-5D responses and the willingness of patients to continue to answer the questionnaire at subsequent clinic visits.

## Methods

### Research design, setting and participants

We conducted a cross-sectional study to assess the feasibility of routinely collecting EQ-5D-5L questionnaires among a convenience sample of women with breast cancer, recruited during clinic visits at the Louise Temerty Breast Cancer Centre (Sunnybrook Health Sciences Centre, Toronto). Participants were eligible for the study if they were: 1) females over 17 years of age; 2) diagnosed with stages I to IV breast cancer; 3) undergoing treatment or active surveillance for their cancer; and 4) English literate. Women were excluded if they could not give written informed consent or complete the study questionnaires on their own. Each woman was asked to participate only once.

The study was designed to interfere as little as possible with the normal clinical workflow. Patients were approached during their scheduled clinic visit, preferably after completing Symptom Screening and Functional Status questionnaires, which are routinely administered in waiting room kiosks. All cancer centres in Ontario have such kiosks (Fig 1).

The Symptom Screening Questionnaire (ESAS, Edmonton Symptom Assessment System) is a self-reported instrument that measures nine common symptoms of cancer: pain, tiredness, drowsiness, nausea, lack of appetite, shortness of breath, depression, anxiety, and well-being [22,23]. ESAS responses are scored on a scale from 0 (no symptoms) to 10 (worst possible symptom). The Patient-Reported Functional Status tool consists of a version of the Eastern Cooperative Oncology Group performance status tool [18]. It was designed to be completed by the patient and asks the patients to rate their activity in the past month from 0 ("normal with no limitation") to 4 ("pretty much bedridden, rarely out of bed"). Both questionnaires take less than 3 minutes to be competed [24,25]. Answers to these questionnaires are reviewed by the clinical care team during the appointment.

Written informed consent was obtained prior to any study procedures. While the answers to the standard of care questionnaires were meant to be discussed with the clinical care team,

---

**Standard of Care**
1) Patients check in at the Main Reception of the Breast Cancer Centre
2) They complete the Symptom Screening and Functional Status questionnaires in the waiting room, near the Main Reception (https://isaac.ontariohealth.ca/)
3) Answers are instantly available in the electronic patient file, to be reviewed by the clinical care team during the visit
4) The data is also available in a provincial database and can be linked to the Cancer Registry and other administrative databases, to monitor symptoms over time and across settings, to improve overall patient care and inform future planning.

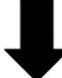

**Study Procedures**
1) Women were recruited in the waiting room of the Breast Cancer Centre, after completing the standard of care questionnaires.
2) Written informed consent was obtained prior to any study procedures.
3) Study questionnaires were self-administered using electronic tablets (iPad®) in the following order:
   - Symptom Screening and Functional Status questionnaires (optional, if not answered before)
   - EQ-5D-5L and willingness question
   - Patient questionnaire (additional demographic and clinical information questions)
4) Data was transferred from the tablets to a temporary databased and then to the research database
5) Steps 3 and 4 required a funcioning wireless internet network
6) Chart review was conducted by study investigators

**Fig 1. Standard of care and study procedures.** Patients were approached for study participation after completing the standard of care questionnaires.

study participants were informed in the written informed consent and orally during recruitment, that EQ-5D-5L answers would not be discussed during the visit or in the future if the questionnaire was also implemented in routine clinical practice. The main objective of EQ-5D collection was to obtain data for drug and health technology reimbursement and decision making.

The study questionnaires were self-administered using electronic tablets (iPad®). Participants could both interrupt answering the study questionnaires at any time and return to answering the questionnaires later during their visit. The recruiters recorded reasons that patients chose not to join the study if they volunteered a reason.

The wireless hospital internet network was used to access the questionnaires on the tablets through a specific web link. We abstracted diagnostic and treatment data from patient charts. All information was stored in the main research database.

## EQ-5D-5L health questionnaire

The EQ-5D-5L questionnaire consists of the EQ-5D-5L descriptive system and the EQ visual analog scale (EQ VAS). EQ-5D-5L has been formally translated and validated into Canadian English and French [26]. The descriptive system includes 5 dimensions: mobility, self-care, usual activities, pain/discomfort, and anxiety/depression. Each dimension has 5 levels: no problems (level 1), slight problems (level 2), moderate problems (level 3), severe problems (level 4), extreme problems (level 5). A unique health state is defined by one level from each of the five dimensions (for example, 11111, if no problems in all 5 dimensions). Each health state is converted into a utility, a cardinal value anchored at 0 (representing death) and 1 (representing full health), by applying a formula that attaches weights to each level in each dimension of the descriptive system [27]. Values below 0, which represent states worse than death, are also possible [28]. We used weights from the Canadian EQ-5D-5L value set estimated by Xie et al [27]. The EQ VAS records the respondents' self-rated health on a vertical, visual analogue scale numbered 0 to 100, where the endpoints are labelled "The best health you can imagine" (100) and "The worst health you can imagine" (0).

The EQ-5D-5L was followed by a question assessing participants' ongoing willingness to answer the questionnaire at subsequent clinic visits. Participants were also asked to provide additional demographic and clinical information.

## Classification into breast cancer states

We classified study participants into five mutually exclusive breast cancer states, based on their disease status at the time of recruitment. Breast cancer states were defined based on a review of the literature, with the intention of being relevant both to clinical practice and economic modeling [17,29]:

1. "First year after primary breast cancer" (State 1): Patients diagnosed with invasive breast cancer (stage I to III) in the 12 months prior to study enrolment and were being treated with curative intent at the time of recruitment or before.

2. "First year after recurrence or new primary breast cancer" (State 2): Patients diagnosed with at least a second metachronous invasive breast cancer or a local-regional recurrence treated with curative intent, in the 12 months prior to study enrolment and without metastatic disease.

3. "Second to fifth year after a primary breast cancer or recurrence treated with curative intent" (State 3): Patients diagnosed with the latest primary invasive breast cancer or a loco-

regional recurrence treated with curative intent more than one year, but fewer than five years, prior to study enrolment and without metastatic disease.

4. "Sixth and following years after a primary breast cancer or recurrence treated with curative intent" (State 4): Patients diagnosed with the latest primary invasive breast cancer or a loco-regional recurrence more than five years prior to study enrolment and without metastatic disease.

5. "Metastatic Breast Cancer" (State 5): Patients with distant or inoperable disease.

## Outcomes

The main outcome was the proportion of participants who fully completed EQ-5D-5L. We also assessed the proportion of eligible patients recruited, the proportion of patients who were ineligible, and the proportion of patients who declined study participation. Other secondary outcomes included: 1) The proportion of participants willing to complete the EQ-5D-5L at each clinic visit; 2) Median completion time for the EQ-5D-5L; 3) Clinical and socio-demographic characteristics associated with willingness to continue to complete the EQ-5D routinely at future clinic visits.

## Sample size

The provincial standard for performance in symptom screening is that each Regional Cancer Centre in Ontario screen 70% of their ambulatory oncology patients at least once per month using the Symptom Screening questionnaire [18]. Based on this standard, we *a priori* defined feasible routine administration of EQ-5D-5L as at least 70% of recruited patients fully completing the questionnaire and defined a fully completed questionnaire as one in which all the items had been answered. We aimed for a sample size of 341 patients, which produced a two-sided 95% confidence interval with a width equal to 10%, when the proportion of patients who fully complete the EQ-5D-5L was 70%.

## Analysis

Descriptive statistics were calculated for all variables and study outcomes. We used the chi-square test or Fisher's exact test for categorical variables to compare characteristics of patients willing to continue to answer EQ-5D-5L routinely and those who were not willing. Multivariable logistic regression was used to identify predictors of willingness to continue to answer EQ-5D-5L routinely. We hypothesised that younger age (<45 years old vs. older), a lower Charlson Comorbidity Index (0 versus higher), higher education level (above 8[th] grade vs. 8[th] grade or lower), native English speaking, and absence of metastatic disease (Health States 1–4 versus 5) would be associated with willingness to continue to answer the EQ-5D-5L routinely. In an exploratory analysis, we tested the hypothesis that "no problems" in each of EQ-5D-5L dimensions would also be associated with willingness to continue to answer EQ-5D-5L routinely. We assessed open-ended questions, such as the reasons offered for non-consent, qualitatively.

Statistical analysis was performed using SAS version 9.4 (SAS Institute, Cary, NC). A two-tailed p-value less than 0.05 was considered statistically significant.

This study was approved by the Sunnybrook's research ethics board (protocol number 2551) and by the University of Toronto Research Ethics Board (protocol number 38199) **and was performed in line with the principles of the Declaration of Helsinki.**

## Results

### Patient recruitment, eligibility, and EQ-5D-5L completion

Between November 2016 and June 2017, we approached 588 women, of whom 341 were eligible and enrolled (recruitment rate: 58%; 95% CI 0.54–0.62), 169 (29%) declined participation and 78 (13%) were ineligible (Fig 2). The most common reason for ineligibility was not being English literate (N = 53; 68%). Reasons for declining study participation were volunteered by 87 women (51%) and are presented in Table 1. The most common reasons were not feeling well enough to participate (N = 30; 34%), lack of time (N = 19; 22%), participation in other studies at the cancer centre (N = 13; 15%), and concerns over privacy or not wanting to provide access to personal health information (N = 12;14%). Only 6 women (7%) cited the use of tablets as a reason for declining participation.

Our primary study outcome was the proportion of fully completed EQ-5D-5L questionnaires by patients recruited; our estimate was 95% (323 of 341 patients recruited; 95% CI 92–97%; Fig 2). This number represented 63% (323/509; 95% CI 59–67%) of all eligible patients approached (Fig 2). Median time for EQ-5D-5L completion was 1.5 minutes (range: 0.35 to 14.7 minutes). Only 2% (N = 5) of the recruited patients were enrolled in a clinical trial at the

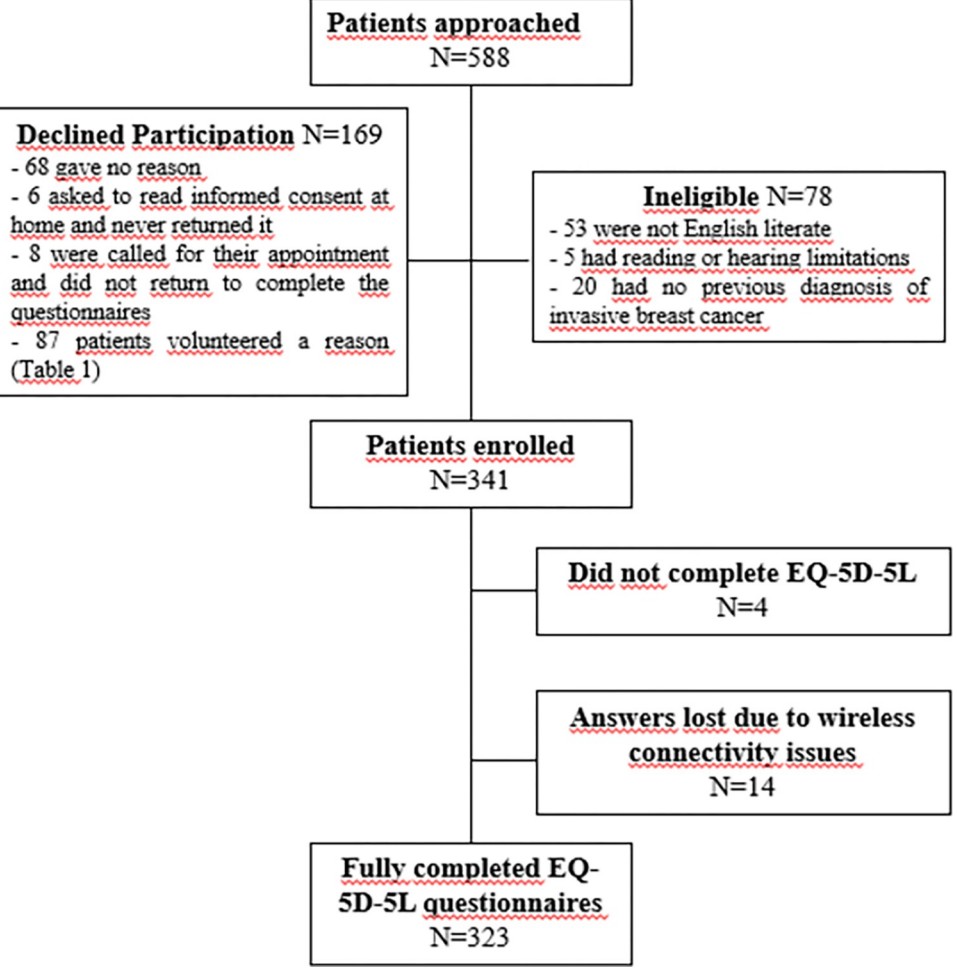

**Fig 2. Patient disposition.** Flow chart summarizing patient recruitment, eligibility, and primary outcome.

Table 1. Reasons given by patients for declining study participation.

| Reasons for declining study participation | Number of Patients (%) |
|---|---|
| Not feeling well enough to participate (too symptomatic or overwhelmed) | 30 (34%) |
| Lack of time to answer the survey | 19 (22%) |
| Participation (currently or in the past) in other studies involving surveys | 13 (15%) |
| Declined participation due to concerns over privacy (feared inappropriate access to private health information or declined chart review) | 12 (14%) |
| Does not like / know how to use tablets | 6 (7%) |
| Does not feel surveys are helpful to herself or other patients and/or preferred to speak with oncologist directly | 6 (7%) |
| Already answered the symptom screening questionnaire | 1 (1%) |
| **Total** | 87 |

time of recruitment. The Symptom Screening and Functional Status questionnaires were completed in the waiting room kiosks by 198 (61%) of the patients who completed EQ-5D-5L, as part of their routine care.

Answers to the EQ-5D-5L were missing for 18 (5%) of the patients enrolled (Fig 2). Of these, 4 (1%) participants were called for their appointments while they were starting to answer the study questionnaires and returned the tablets without answering any questions. Answers of the other 14 (4%) patients were lost during data transfer to the temporary database (either because the tablets were inactive for more than one hour and the system logged out for privacy reasons or because of interruptions in the wireless network). All data were successfully transferred from the temporary database to the main research database.

## Patient characteristics

Participants who fully answered the EQ-5D-5L had a mean age of 58 years, and most were White (N = 172; 53%), born outside Canada (N = 182; 57%), and had a high level of education (N = 182; 56% attended or graduated college or university; N = 73; 23% had postgraduate or professional degrees) (Table 2). Although all participants were English literate, 35% (N = 115) primarily spoke other languages at home. Most participants (N = 228; 71%) had been diagnosed with breast cancer in the 7 years prior to enrollment and were receiving systemic treatment (N = 230; 71%). After we classified participants into our five mutually exclusive breast cancer states, only 5 were included in Health State 2 ("First Year after recurrence or new primary breast cancer") (Table 2).

## EQ-5D-5L results

Across all health states, most participants answered the EQ-5D-5L questions reporting "No problems" (level 1) for "Mobility" and "Self-Care" (Table 3). Most patients in State 4, who were further away from their breast cancer diagnosis and had been treated with curative intent, reported "No Problems" (level 1) for "Usual Activity", "Pain/Discomfort" and "Anxiety/ Depression" dimensions, while participants in the other States reported "Slight" to "Moderate" problems (levels 2 and 3, respectively). Very few participants reported "Severe" to "Extreme" problems (levels 4 and 5, respectively) in any dimension. As expected, patients with metastatic breast cancer (State 5) reported the greatest number of problems.

The mean EQ-5D-5L utility value for all breast cancer patients was 0.83 (SD 0.13), with the highest values (0.86; SD 0.14) in State 4 and the lowest values (0.78; SD 0.14) in patients with State 5 (Table 4). The mean EQ-5D-5L VAS value (Table 4) for all breast cancer patients was

**Table 2. Characteristics of the patients who completed the EQ-5D-5L.**

| Characteristics | N = 323 (%) |
|---|---|
| Age (Mean; SD; min-max) | 58 (SD:12;28–90) |
| **Age** | |
| < 45 years | 38 (12%) |
| 45–64 years | 192 (59%) |
| ≥65 years | 93 (29%) |
| **Menopausal Status** | |
| Pre-menopausal | 51 (16%) |
| Post-menopausal | 227 (70%) |
| **Marital Status** | |
| Married or common-law | 225 (70%) |
| Separated, divorced, or widowed | 69 (21%) |
| Single or never married | 27 (8%) |
| **Education** | |
| ≤ Grade 8 | 11 (3%) |
| Attended or graduated high school | 56 (17%) |
| Attended or graduated college or university | 182 (56%) |
| Postgraduate or professional degree | 73 (23%) |
| **Employment Status** | |
| Retired | 103 (32%) |
| Unemployed | 28 (9%) |
| Working full-time or part-time | 139 (43%) |
| Other (e.g, on leave, disability) | 51 (16%) |
| **Annual Family Income** | |
| <$30,000 | 37 (11%) |
| $30,000 to $59,999 | 58 (18%) |
| $60,000 to $89,999 | 51 (16%) |
| $90,000 to $119,999 | 26 (8%) |
| $120,000 to $149,999 | 27 (8%) |
| >$150,000 | 47 (15%) |
| **Born in Canada** | |
| Yes | 129 (40%) |
| No | 183 (57%) |
| **Primary Language spoken at home** | |
| English | 205 (63%) |
| Other | 115 (36%) |
| **Racial or ethnic group** | |
| Asian—East (eg, Chinese, Japanese, Korean) | 45 (14%) |
| Asian—South (eg, Indian, Pakistani, Sri Lankan) | 22 (7%) |
| Asian—South East (eg, Malaysian, Filipino, Vietnamese) | 22 (7%) |
| Black—Caribbean (eg, Barbadian, Jamaican) | 8 (2%) |
| Indian—Caribbean (eg, Guyanese with origins in India) | 8 (2%) |
| Latin American (eg, Argentinean, Chilean, Salvadoran) | 9 (3%) |
| Middle Eastern (eg, Egyptian, Iranian, Lebanese) | 26 (8%) |
| Mixed heritage (eg, Black—African and White—North American) | 3 (1%) |
| White—European (eg, English, Italian, Portuguese, Russian) | 65 (20%) |
| White—North American (eg, Canadian, American) | 107 (33%) |
| **Charlson Comorbidity Index** | |
| 0 | 221 (68%) |
| 1–2 | 85 (26%) |
| ≥3 | 17 (5%) |
| **Breast Cancer stage at diagnosis** | |
| Stage I | 100 (31%) |
| Stage II | 142 (44%) |
| Stage III | 61 (19%) |
| Stage IV | 16 (5%) |

*(Continued)*

**Table 2.** (Continued)

| Characteristics | N = 323 (%) |
|---|---|
| **Breast Cancer Subtype** | |
| Hormone-Receptor Positive | 223 (69%) |
| HER2 Positive | 55 (17%) |
| Triple negative | 40 (12%) |
| **Breast cancer Surgery** | 132 (41%) |
| Mastectomy | 159 (49%) |
| Lumpectomy | 169 (52%) |
| SLNB | 118 (37%) |
| ALND | |
| **Current treatment** | |
| Chemotherapy (+/- targeted therapy) | 48 (15%) |
| Hormonal treatment (+/- targeted therapy) | 170 (53%) |
| Targeted therapy (only) | 12 (4%) |
| Radiotherapy | 2 (1%) |
| **Breast Cancer State** | |
| First year after primary breast cancer | 78 (24%) |
| First year after recurrence or new primary breast cancer | 5 (2%) |
| Second to fifth year after primary breast cancer or recurrence | 104 (32%) |
| Sixth and following years after primary breast cancer or recurrence | 55 (17%) |
| Metastatic Breast Cancer | 81 (25%) |

N, number; SD, standard deviation; Min, minimum; Max, maximum; SLNB, sentinel lymph node biopsy; ALND, axillary lymph node dissection.

75.7 (SD 17.45), with the highest value -also for State 4 (81.7; SD14.0) and the lowest for State 5 (68.1; SD 19.4).

## Willingness to complete EQ-5D-5L routinely

Of participants who completed EQ-5D-5L, 135 (42%) said they would definitely continue to answer EQ-5D-5L routinely at subsequent clinic visits, 117 (36%) were very likely to do so, 42 (13%) were unsure, 18 (6%) were very unlikely, and 10 (3%) would definitely not (Table 5). Reasons for choosing one of the latter three options were provided by 32 participants (46%) (S1 Table) and frequently included: the time consuming/repetitive nature of the questions; taking time away from their appointment; not seeing the benefit of answering the questionnaire; indicating that the questions were not appropriate for their situation; stating that the EQ-5D questionnaire was too similar to the symptom screening questionnaire; not being symptomatic in most visits or being healthy; and preferring to discuss symptoms with as oncologist.

In the unadjusted analysis (S2 Table), the Charlson Comorbidity Index and the educational level were associated with willingness to answer EQ-5D routinely, while any level of reported "problems" in the EQ-5D-5L mobility dimension was associated with unwillingness to answer EQ-5D-5L routinely. In multivariable analysis (Table 6), patients with a Charlson comorbidity index of 1 to 2 (OR 0.77, 95% CI 0.40–1.49) and ≥ 3 (OR 0.25, 95% CI 0.08–0.73) were less willing to answer EQ-5D-5L routinely (joint p = 0.039) compared with patients with an index value of 0, and patients with an education level of high school (OR 11.98, 95% CI 2.60–55.29) or higher (College / University OR 10.95, 95% CI 2.61–45.90 and Postgraduated/ Professional OR 16.55, 95% CI 3.59–76.37) more willing to answer EQ-5D-5L routinely than patients with an eighth grade education or less (joint p = 0.004). In the exploratory analysis that added problems/ no problems in each of the EQ-5D-5L dimensions to the multivariable model, only the level of education continued to be significantly associated with willingness to continue to answer EQ-5D-5L at subsequent visits (S3 Table).

**Table 3. EQ-5D-5L health profile by breast cancer state.**

| EQ-5D Dimensions | | Breast Cancer States (N; percent of column total) | | | | | Total (N = 323) |
|---|---|---|---|---|---|---|---|
| | | State 1 (N = 78) | State 2 (N = 5) | State 3 (N = 104) | State 4 (N = 55) | State 5 (N = 81) | |
| **Mobility** | No problems | 61; 78% | 3; 60% | 72; 69% | 42; 76% | 42; 52% | 220; 68% |
| | Slight problems | 11; 14% | 2; 40% | 22; 21% | 8; 15% | 25; 31% | 68; 21% |
| | Moderate problems | 6; 8% | - | 9; 9% | 2; 4% | 10; 12% | 27; 8% |
| | Severe problems | - | - | 1;1% | 2; 4% | 3; 4% | 6; 2% |
| | Extreme problems | - | - | - | 1; 2% | 1; 1% | 2; 1% |
| **Self-Care** | No problems | 66; 85% | 3; 60% | 92; 88% | 51; 93% | 63; 78% | 275; 85% |
| | Slight problems | 10; 13% | 2; 40% | 10; 10% | 3; 5% | 13; 16% | 38; 12% |
| | Moderate problems | 1; 1% | - | 2; 2% | 1; 2% | 4; 5% | 8; 2% |
| | Severe problems | - | - | - | - | - | - |
| | Extreme problems | 1; 1% | - | - | - | 1; 1% | 2; 1% |
| **Usual Activity** | No problems | 31; 40% | 1; 20% | 59; 57% | 44; 80% | 28; 35% | 163; 50% |
| | Slight problems | 34; 44% | 2; 40% | 33; 32% | 5; 9% | 35; 43% | 109; 34% |
| | Moderate problems | 11; 14% | 2; 40% | 11; 11% | 6; 11% | 15; 19% | 45; 14% |
| | Severe problems | 1; 1% | - | 1; 1% | - | 1; 1% | 3; 1% |
| | Extreme problems | 1; 1% | - | - | - | 2; 2% | 3; 1% |
| **Pain/ Discomfort** | No problems | 38; 49% | 1; 20% | 31; 30% | 33; 60% | 20; 25% | 123; 38% |
| | Slight problems | 28; 36% | 2; 40% | 51; 49% | 12; 22% | 33; 41% | 126; 39% |
| | Moderate problems | 10; 13% | 2; 40% | 16; 15% | 8; 15% | 23; 28% | 59; 18% |
| | Severe problems | 1; 1% | - | 6; 6% | 1; 2% | 4; 5% | 12; 4% |
| | Extreme problems | 1; 1% | - | - | 1; 2% | 1; 1% | 3; 1% |
| **Anxiety/ Depression** | No problems | 35; 45% | 3; 60% | 45; 43% | 28; 51% | 24; 30% | 135; 42% |
| | Slight problems | 30; 38% | 1; 20% | 44; 42% | 18; 33% | 37; 46% | 130; 40% |
| | Moderate problems | 11; 14% | 1; 20% | 14; 13% | 8; 15% | 19; 23% | 53; 16% |
| | Severe problems | 1; 1% | - | 1; 1% | 1; 2% | 1; 1% | 4; 1% |
| | Extreme problems | 1; 1% | - | - | - | - | 1; 0.3% |

State 1, "First year after primary breast cancer"; State 2, "First year after recurrence or new primary breast cancer"; State 3, "Second to fifth year after a primary breast cancer or recurrence treated with curative intent"; State 4, "Sixth and following years after a primary breast cancer or recurrence treated with curative intent"; State 5, "Metastatic Breast Cancer".

**Table 4. EQ-5D-5L utility and VAS values by breast cancer state.**

| EQ-5D-5L | Breast Cancer State | N | % | Mean | SD | Median | IQR | Min | Max |
|---|---|---|---|---|---|---|---|---|---|
| Index Values | State 1 | 78 | 24 | 0.84 | 0.14 | 0.87 | 0.82–0.93 | 0.23 | 0.95 |
| | State 2 | 5 | 2 | 0.82 | 0.09 | 0.78 | 0.76–0.87 | 0.73 | 0.95 |
| | State 3 | 104 | 32 | 0.84 | 0.12 | 0.86 | 0.81–0.91 | 0.21 | 0.95 |
| | State 4 | 55 | 17 | 0.86 | 0.14 | 0.91 | 0.85–0.95 | 0.36 | 0.95 |
| | State 5 | 81 | 25 | 0.78 | 0.14 | 0.81 | 0.70–0.89 | 0.29 | 0.95 |
| | Total | 323 | 100 | 0.83 | 0.13 | 0.87 | 0.79–0.91 | 0.21 | 0.95 |
| VAS values | State 1 | 78 | 24 | 76.0 | 18.7 | 80.0 | 61–90 | 10 | 100 |
| | State 2 | 5 | 2 | 82.4 | 12.3 | 80.0 | 73–89 | 70 | 100 |
| | State 3 | 104 | 32 | 77.8 | 14.7 | 80;5 | 69–90 | 36 | 100 |
| | State 4 | 55 | 17 | 81.7 | 14.0 | 82.0 | 74–94 | 49 | 100 |
| | State 5 | 81 | 25 | 68.1 | 19.4 | 70.0 | 52–84 | 21 | 100 |
| | Total | 323 | 100 | 75.7 | 17.5 | 80.0 | 62–90 | 10 | 100 |

State 1, "First year after primary breast cancer"; State 2, "First year after recurrence or new primary breast cancer"; State 3, "Second to fifth year after a primary breast cancer or recurrence treated with curative intent"; State 4, "Sixth and following years after a primary breast cancer or recurrence treated with curative intent"; State 5, "Metastatic Breast Cancer"; N, number; SD, standard deviation; IQR, intra-quartile range; Min, minimum; Max, maximum.

**Table 5. Willingness to continue to answer EQ-5D-5L routinely at subsequent clinic visits.**

| Willingness to answer EQ-5D-5L | Breast Cancer States (N; percentage of column total) | | | | | Total (N = 323) |
| --- | --- | --- | --- | --- | --- | --- |
| | State 1 (N = 78) | State 2 (N = 5) | State 3 (N = 104) | State 4 (N = 55) | State 5 (N = 81) | |
| **Definitely** | 34; 44% | - | 49; 47% | 23; 42% | 29;36% | 135; 42% |
| **Very likely** | 27; 35% | 3; 60% | 36; 35% | 22; 40% | 29; 36% | 117; 36% |
| Total | 61; 78% | 3; 60% | 85; 82% | 45; 82% | 58; 72% | 252; 78% |
| **Unsure** | 11; 14% | - | 13; 13% | 7; 13% | 11; 14% | 42; 13% |
| **Very unlikely** | 5; 6% | 1; 20% | 3; 3% | 1; 2% | 8; 10% | 18; 6% |
| **Definitely not** | - | 1; 20% | 3; 3% | 2; 4% | 4; 5% | 10; 3% |
| Total | 5; 6% | 2; 40% | 6; 6% | 3; 6% | 12; 15% | 28; 9% |

Participants were asked if they were willing to continue to answer EQ-5D-5L routinely, at subsequent clinic visits, if asked to do so. Possible answers were "Definitely", "Very likely", "Unsure", "Very unlikely", "Definitely not". Patients who chose on of the latter 3 options, were asked to volunteer a reason for their answer. State 1, "First year after primary breast cancer"; State 2, "First year after recurrence or new primary breast cancer"; State 3, "Second to fifth year after a primary breast cancer or recurrence treated with curative intent"; State 4, "Sixth and following years after a primary breast cancer or recurrence treated with curative intent"; State 5, "Metastatic Breast Cancer"; N, number.

**Table 6. Characteristics associated with willingness to continue to answer EQ-5D at each clinic visit.**

| Characteristics | Willing N = 252; 78% | Unsure / Unwilling N = 70; 22% | Adjusted OR (95% CI) | p-value |
| --- | --- | --- | --- | --- |
| **Age** | | | | 0.555 |
| < 45 years | 29; 76% | 9; 24% | reference | |
| 45–64 years | 152; 80% | 39; 20% | 1.59 (0.66–3.84) | |
| ≥ 65 years | 71; 76% | 22; 24% | 1.66 (0.61–4.52) | |
| **Charlson Comorbidity Index** | | | | 0.039 |
| 0 | 179; 81% | 41; 19% | reference | |
| 1–2 | 64; 75% | 21; 25% | 0.77 (0.40–1.49) | |
| ≥ 3 | 9; 53% | 8; 47% | 0.25 (0.08–0.73) | |
| **Education** | | | | 0.004 |
| Grade 8 education or less | 3; 27% | 8; 73% | reference | |
| Some of completed High school | 45;80% | 11; 20% | 11.98 (2.60–55.29) | |
| Some of completed college or university | 142; 78% | 40; 22% | 10.95 (2.61–45.90) | |
| Some of completed postgraduate / professional | 62; 85% | 11; 15% | 16.55 (3.59–76.37) | |
| **Primary Language** | | | | 0.388 |
| English | 163; 80% | 42; 20% | reference | |
| Other | 87; 76% | 28; 24% | 0.77 (0.43–1.38) | |
| **Breast CancerState** | | | | 0.429 |
| State 1 | 61; 79% | 16; 21% | 1.76 (0.79–3.90) | |
| State 2 | 3; 60% | 2; 40% | 0.55 (0.08–3.61) | |
| State 3 | 85; 82% | 19; 18% | 1.73 (0.84–3.57) | |
| State 4 | 45; 82% | 10; 18% | 1.50 (0.62–3.64) | |
| State 5 | 58; 72% | 23; 28% | reference | |

We hypothesised that younger patients (younger than 45 years old), with a lower Charlson Comorbidity Index (0 versus higher), higher education level (versus ≤ 8 grade), English native speakers and who did not present metastatic disease (States 1–4 versus 5) would be more willing to continue to answer the EQ-5D-5L routinely. OR, odds ratio.

## Discussion

We found that routine administration of EQ-5D-5L for breast cancer patients is feasible in a clinical care setting where patients routinely answer PROs using a web-based system, with 95% of patients fully completing EQ-5D-5L. The distribution of breast cancer staging and sub-type in our study was typical of that described in the literature, but our pre-specified breast cancer states had limitations since we only recruited 5 patients to breast cancer state 2 [30].

The median time for EQ-5D-5L completion was acceptable and most participants were willing to complete EQ-5D-5L at subsequent clinic visits. Reasons given by participants for unwillingness to answer the questionnaire at subsequent visits included the overlap between some questions of EQ-5D-5L and the symptom screening questionnaire, not seeing the benefit of answering the questionnaire and indicating that the questions were not appropriate for their situation. Addressing these reasons is important for increasing patient participation and retention. The use of tablets was well-accepted and did not seem to interfere with study participation, although these findings might not be generalizable to all clinical settings, since patients in Ontario are used to answer PROs in electronic format.

However, the number of patients who fully completed EQ-5D-5L represents only 58% of presumed eligible patients approached for study participation. Although we were not able to determine if there were significant differences between study participants and non-participants, we found that participants were less willing to routinely answer the EQ-5D-5L if they had comorbidities or a lower education level. These results are exploratory and need confirmation, because the model was prone to overfitting because it had many predictors.

The mean EQ-5D-5L utility and VAS values obtained for all breast cancer patients were high (0.83 and 75.7, respectively). These results were lower than the normative values for EQ-5D-5L found for Canadian women, which were respectively 0.869 and 82.8, and close to the values reported for Canadian women with chronic health conditions (utility index 0.841 and VAS value 79.3) [31]. The highest values in our study, which are similar to "healthy" Canadian women, were observed for patients who had breast cancer treated with curative intent more than six years prior to enrolment, and the lowest for patients with metastatic disease Surprisingly, our patients reported "no problems" or a low level of "problems" in most EQ-5D-5L dimensions, even if we consider patients with metastatic disease, which raises the question of whether EQ-5D adequately captures the health status of these patients or whether our study population was representative of all breast cancer patients. The dimensions in which a higher level of problems was reported were "Usual activities" and "Anxiety / Depression".

Our findings are comparable to those of other studies that have incorporated the collection of PRO questionnaires into the routine care of cancer patients. According to the Ontario Cancer System Quality Index, the screening rate for the Symptom Screening and Functional Status questionnaires for breast cancer patients, which are collected without the need of a consent form and are used in direct clinical care, was 53% in 2018. Population-based initiatives without a clinical care component that collect PRO from cancer survivors using clinical staff to obtain informed consent [32], have consent rates ranging from 61% (in the electronic Patient-reported Outcomes from Cancer Survivors (ePOCS) system) [26,33,34] to 95% (in the Cancer of the Prostate Strategic Urologic Research Endeavour (CaPSURE) database) [35–37] Within the ePOCS system, which also uses a web-based system for electronic capture of different PROs, including the EQ-5D, a decrease in questionnaire completion rates was observed over time, with 85% of recruited patients completing the questionnaires at baseline and 66% at 15 months, although 86% of recruited patients said they would be willing to use the system long-term [26]. We hypothesize that the cross-sectional nature of our study, as well as the explanation from study staff about why data was being collected, might explain the high completion

rate for EQ-5D-5L. Participant fatigue might decrease participation over time in clinics that implement routine collection of the EQ-5D-5L. Alternatively, collection without informed consent may increase participation over what we have observed, although our study was very pragmatic apart from this requirement. The net effect of these factors is a topic for future study.

Our study has limitations. We recruited patients from an outpatient clinic of a single tertiary cancer centre, and although our study population is representative of the patients followed at our centre, these patients might not be representative of breast cancer patients in other settings. Patients recruited were mostly young, White, and well-educated. We did not include patients with more severe disease (patients admitted to hospital or to palliative care, for example) or long-term survivors who were no longer followed at the cancer centre. We excluded patients who were not English literate. Our study may also have been prone to non-response bias, as we included patients who were feeling relatively well, as patients reported feeling ill as the main reason for not participating, including being too symptomatic and too overwhelmed (for example, because they were waiting for a diagnosis or exam results).

Our findings can inform future initiatives regarding routine collection of PROs. Asking patients to complete EQ-5D-5L, the Symptom Screening and the Functional Status questionnaires together, at the time of their scheduled appointments, is an opportunity for patients to get acquainted with the web-based system and to collect data at important times in each patients' cancer trajectory (e.g., diagnosis, start of a new treatment, progression of disease). The EQ-5D-5L is appealing to add as a routinely collected PRO because it is a short questionnaire that can be easily answered outside the clinic, in a computer, or on a smart phone, a few days before or after the clinic appointment, and even after patients are discharged from the cancer clinic, improving patient convenience, and potentially, participation and retention. The possibility of answering EQ-5D outside cancer clinics is especially relevant to capture data on quality-of-life over time and to include long-term survivors or patients at the end of life. In the future, a system of automatic electronic reminders could be set up at specified time points, to investigate if patient participation would increase.

Although the EQ-5D can be used in individual patient consultations, for example for supporting clinical decisions or for shared decision making, our primary goal with this study was not patient care but to use the collected data to inform drug and health technology reimbursement and decision making [38]. Providing patients with practical examples of how the data is used, and even explore possible applications of the data collected in clinical practice, might possibly increase their willingness to answer EQ-5D routinely [32]. Routinely collected EQ-5D data is already used to inform patient-provider decision making in other areas, such as knee or hip replacement surgery [39,40]. The clinical usefulness of integrating patient-reported outcome measures (including the Edmonton Symptom Assessment System- revised: Renal-ESAS-r and the EQ-5D-5L) in the clinical management of hemodialysis patients is being assessed in a randomized clinical trial in Alberta and Ontario (NCT03535922). Similar research is lacking in cancer patients.

## Conclusions

In conclusion, our study demonstrated that tablet-based collection of EQ-5D-5L in the context of routine clinical practice was feasible for breast cancer patients. EQ-5D-5L can be easily administered, in addition to the Symptom Screening and Functional Status questionnaires. However, many patients declined study participation or reported being in full health, raising concerns about whether this method of collecting EQ-5D adequately represents the health status of all breast cancer patients.

## Supporting information

**S1 Table. Reasons given by study participants for not wanting to answer EQ-5D-5L at subsequent clinic visits.**
(PDF)

**S2 Table. Characteristics associated with willingness to continue to answer EQ-5D at each clinic visit, unadjusted analysis.**
(PDF)

**S3 Table. Characteristics associated with willingness to continue to answer EQ-5D at each clinic visit, exploratory analysis.**
(PDF)

## Author Contributions

**Conceptualization:** Sofia Torres, Ahmed M. Bayoumi, Ana B. K. Abrahao, Maureen Trudeau, Kathleen I. Pritchard, Chun Nim Li, Nicholas Mitsakakis, Geoffrey Liu, Murray Krahn.

**Data curation:** Sofia Torres, Ana B. K. Abrahao, Maureen Trudeau, Kathleen I. Pritchard, Chun Nim Li, Nicholas Mitsakakis, Murray Krahn.

**Formal analysis:** Sofia Torres, Ahmed M. Bayoumi, Ana B. K. Abrahao, Nicholas Mitsakakis, Murray Krahn.

**Investigation:** Sofia Torres, Ahmed M. Bayoumi, Ana B. K. Abrahao, Maureen Trudeau, Kathleen I. Pritchard, Nicholas Mitsakakis, Geoffrey Liu, Murray Krahn.

**Methodology:** Sofia Torres, Ahmed M. Bayoumi, Ana B. K. Abrahao, Maureen Trudeau, Kathleen I. Pritchard, Nicholas Mitsakakis, Geoffrey Liu, Murray Krahn.

**Project administration:** Sofia Torres, Ahmed M. Bayoumi, Ana B. K. Abrahao, Maureen Trudeau, Chun Nim Li, Murray Krahn.

**Resources:** Sofia Torres, Ana B. K. Abrahao, Maureen Trudeau, Kathleen I. Pritchard, Chun Nim Li, Murray Krahn.

**Software:** Sofia Torres, Chun Nim Li, Murray Krahn.

**Supervision:** Sofia Torres, Ahmed M. Bayoumi, Ana B. K. Abrahao, Maureen Trudeau, Kathleen I. Pritchard, Nicholas Mitsakakis, Geoffrey Liu, Murray Krahn.

**Validation:** Sofia Torres, Ahmed M. Bayoumi, Maureen Trudeau, Nicholas Mitsakakis, Geoffrey Liu, Murray Krahn.

**Visualization:** Sofia Torres, Murray Krahn.

**Writing – original draft:** Sofia Torres, Murray Krahn.

**Writing – review & editing:** Sofia Torres, Ahmed M. Bayoumi, Ana B. K. Abrahao, Maureen Trudeau, Kathleen I. Pritchard, Chun Nim Li, Nicholas Mitsakakis, Geoffrey Liu, Murray Krahn.

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
