## [Decision Letter · Decision Letter 0]

22 Aug 2023

PONE-D-23-18133Feasibility of routine collection of health state utilities using EQ-5D-5L in a breast cancer outpatient clinicPLOS ONE

Dear Dr. Torres,

Thank you for submitting your manuscript to PLOS ONE. After careful consideration, we feel that it has merit but does not fully meet PLOS ONE’s publication criteria as it currently stands. Therefore, we invite you to submit a revised version of the manuscript that addresses the points raised during the review process.

We look forward to receiving your revised manuscript.

Kind regards,

Elisa Ambrosi

Academic Editor

PLOS ONE

Reviewers' comments:

Reviewer's Responses to Questions

**Comments to the Author**

1. Is the manuscript technically sound, and do the data support the conclusions?

Reviewer #1: Partly

Reviewer #2: Partly

2. Has the statistical analysis been performed appropriately and rigorously? 

Reviewer #1: I Don't Know

Reviewer #2: No

3. Have the authors made all data underlying the findings in their manuscript fully available?

Reviewer #1: Yes

Reviewer #2: No

4. Is the manuscript presented in an intelligible fashion and written in standard English?

Reviewer #1: Yes

Reviewer #2: Yes

5. Review Comments to the Author

Reviewer #1: Abstract – What does this actually mean “Answers were linked to diagnostic and treatment data.” Please revise the writing to make it clear.

There are similar studies, please review them and describe the similiarties and differences between the existing studies and the present study:

https://hqlo.biomedcentral.com/articles/10.1186/s12955-022-02047-0

https://www.thieme-connect.com/products/ejournals/html/10.1055/s-0043-116223

https://www.sciencedirect.com/science/article/pii/S2212109918300487

Are there results about the characteristics of patients who did not fully complete the EQ-5D-5L? Such information can help us understand why patients did not use it.

I see that the authors have published a similar paper, what are the similarities and differences? https://www.annalsofoncology.org/article/S0923-7534(20)38390-3/fulltext

Do the authors think that the use of tablet computers affected the use of the EQ-5D-5L?

Would it be possible that willingness to complete the instrument may be affected not only by the EQ-5D-5L but also by computer use? Those patients who did not complete it because of the computer?

The discussion should be significantly improved. Implications for practice and policy can be included.

What does this mean “We excluded patients based on literacy.”?

A conclusion section is missing. Please provide a conclusion section,

I suggest to use a passive voice for the first sentence of the paper, for example, “A cross-sectional study was conducted to investigate the feasibility …”

Reviewer #2: If feasibility was the sole objective of this study then one might be forgiven for wondering about the need for a research project; completion rate and/or willingness to complete alone in an observational setting would yield an answer. But this manuscript goes well beyond feasibility and provides descriptive material that speaks to the performance of EQ-5D-5L in ambulatory patients in a routine hospital clinic setting. It is way more that “collecting utilities”. The title could really better reflect this significance.

The rationale for collecting utilities is restricted exclusively to their use in economic evaluation. QALYs are not (and possibly never will be) used to make operational decisions in a clinical setting about individual patients.

P4-Line79 refers to EQ-5D as “the most widely used preference-based HRQOL measure”. This is incorrect. EQ-5D is fundamentally a generic measure of health status (aka HrQoL) defined by a classification system that describes health states. These states can be scored/valued/weighted in several different ways – ONE of which is to apply social preference weights. In that highly restricted form alone (ie suitable for QALY computation) it is indeed preference-based. However, EQ-VAS which sits alongside the self-reported EQ-5D classifier, represents a wholly different perspective (the patient themselves) and expresses a value for their health on a 0-100 metric that has absolutely no relationship to the weighted index form. Zero on the EQ-VAS indicates worst possible health status – not “dead”.

To be honest, the Lines 66 -72 could easily be omitted without loss. Some of the sentiments might be accommodated within the discussion section.

The final paragraph on page 4 is better placed in the Methods section on the next page.

In re-reading the introduction the emphasis on economic applications skews the potential usefulness of this paper. Start with the patient and ask yourself which is more relevant to YOU – your self-reported health status on the 5 dimensions and your EQ-VAS, or the social preference-weighted index based on the general population’s views about YOUR health.

Line 96 “Symptom Screening and Functional Status questionnaires in waiting room kiosks”

These instruments may have worldwide recognition but I was unable to find any reference in the citations or on the web – a brief description would help. The reason? If we are using completion rates then we need to know more about the context and presentation – are waiting room kiosks a standard feature of Canadian hospitals as they are unknown to this reviewer. How long on average does it take to complete these standard questionnaires? What was the order of presentation – was it randomised? Discontinuation of EQ-5D through fatigue/boredom after a 10-minute slog through preceding questionnaires would be a different matter than rejecting it when it was offered first.

Only 61% of patients completing EQ-5D also completed the screening questionnaires (Line 194) so it would be helpful to know more about the precise circumstances under which EQ-5D was completed. How was the timing of EQ-5D completion carried out?

Line 120 questionnaire at each clinic visit. Does this mean at subsequent clinic visits? In which case this could be a good proxy for “satisfaction” with EQ-5D (or acceptance of it) – a positive performance attribute.

The classification into health “states” according to disease severity at recruitment is a complication. EQ-5D defines health states, so why not use the term disease severity for the patient’s baseline clinically-assessed health status? In point of fact isn’t this classification better thought of as an index of chronicity – time since initial diagnosis?

Was all clinically relevant information on patients tested for its association with completion/satisfaction - might the patient's "journey" / treatment pathway be associated with EQ-5D performance? Supplementary free text includes interesting clues suggestive of further investigation - if not by the authors then by others. It would be really good to promote some of this verbatim material to the main text and to have some of its implications explored in the discussion. Surely this is mainstream to the examination of feasibility of routine HrQoL more generally - hence its wider value here.

Line 151 How was the 70% threshold for completion feasibility established, rule of thumb or external reference? The references cited in the discussion maybe?

Analysis with such rich patient data requires careful management. The seeming rush to report utilities should not force out the fundamental material that has been relegated to supplementary tables. Table S2 indicates 2 issues – firstly the classification of patients by disease severity is not helpful as it stands with n=5 for HS2; secondly for the evident high rates of no problem responses for 3/5 dimensions in the survey sample as a whole. The reporting of EQ-5D in its index form effectively conceals this facet of its performance.

If patient completion rates and degree of acceptance/satisfaction are the primary indicators in this study, then why not exploit the EQ-5D classifier (dichotomize problems none/any) and test whether self-reported problems (at any level) are likely to reduce “performance”? If such a non-random influence were apparent then there are clear implications !!

If both index and EQ-VAS are to be reported, then it seems reasonable to include some assessment of their relative performance. Does one work better than the other? Of course, the index is totally dependent on the self-classifier. Ultimately, what's the point in establishing "feasibility" if performance as a METRIC is not well-established.

Given that there are now published population reference “norms” for Canada (Yan, J., Xie, S., Johnson, J.A. et al. Canada population norms for the EQ-5D-5L. Eur J Health Econ (2023). https://doi.org/10.1007/s10198-023-01570-1), it would be especially important to know how patients in this study compare. The study population average index reported here is 0.86; the mean population average for women in Yan et al is 0.869. Taken at face value, this would suggest that the women in this study are (on average) at least as “healthy” as their Canadian peers.

If, as indicated in the discussion, the primary goal of the study is to collect data for drug and health technology reimbursement and decision making, then the capacity of EQ-5D to adequately discriminate between known groups is important. One must wonder what patients think about when completing (any) routine assessment of this type – if they were TOLD that their responses would be used to inform HTA decision-makers, would they still agree/complete? Should they be so informed?

Feasibility testing obviously has some value, but ultimately all stakeholders need to have confidence in the metrics that we use to measure health “benefits”. This study has wider potential value in shaping/informing the scientific community and society at large.

6. PLOS authors have the option to publish the peer review history of their article (what does this mean?). If published, this will include your full peer review and any attached files.

Reviewer #1: No

Reviewer #2: **Yes: **Paul Kind

---

## [Author Response · Author response to Decision Letter 0]

6 Oct 2023

Sofia Torres

sofia.cidtorres@mail.utoronto.ca

10 September 2023

Elisa Ambrosi, PhD, Academic Editor

Plos One

Dear Dr. Ambrosi and reviewers, 

On behalf of my colleagues, as corresponding author, I would like to thank the editor and external reviewers for the thoughtful review of our manuscript entitled “Feasibility of routine collection of health state utilities using EQ-5D-5L in a breast cancer outpatient clinic”.

The paper was reviewed, and all comments/ suggestions made by both reviewers addressed. All the following alterations are tracked and explained on the reviewed manuscript:

Reviewer 1:

# Comment 1: Abstract – What does this actually mean “Answers were linked to diagnostic and treatment data.” Please revise the writing to make it clear.

Answer to comment 1: Sentence was changed to: “Diagnostic and treatment data were abstracted from patient charts.”

# Comment 2: There are similar studies, please review them and describe the similiarties and differences between the existing studies and the present study:

https://hqlo.biomedcentral.com/articles/10.1186/s12955-022-02047-0

https://www.thieme-connect.com/products/ejournals/html/10.1055/s-0043-116223

https://www.sciencedirect.com/science/article/pii/S2212109918300487

Answer to comment 2: Our discussion already included several studies we considered relevant for the discussion (lines 399 to 416 of the revised manuscript). The studies mentioned by the reviewer have significant differences in terms of objectives, methodology, population included and setting, compared with our study. 

The first study was a longitudinal study that included patients with musculoskeletal problems in primary care clinics. Patients with an active cancer diagnosis were excluded. Trained research assistants helped the patients with technical questions and reading out the questions if necessary. EQ-5D results were given to the patients to be reviewed and problems addressed by trained doctors during consultations. Both patients´ and doctors’ perspectives were assessed. Thus, both the administration of the EQ-5D and the purpose of assessment differed markedly from those in our study.

The second study included 17 patients with advanced breast cancer and assessed the feasibility of incorporating three electronic questionnaires (one of which EQ-5D) in a cancer registry. The aims of the study were to assess the feasibility of using dedicated study staff (clinical trial unit trained staff, resident physicians treating the patients and nurses involved in cancer care) to initiate patient data entry (account step up took about 13 minutes); to investigate the patients’ views on the feasibility of the process, including preference for paper versus electronic questionnaires); and to assess the quality of the patient-reported outcome data. Again, the questions addressed by this study (account setup, paper vs. electronic) make it sufficiently dissimilar that we do not feel that we need to include it. 

The third study assesses the measurement equivalence of the original paper version of EQ-5D and an adapted tablet version in the general Brazilian population (Portuguese version of EQ-5D) but was not focused on feasibility, which was our main outcome measure.

# Comment 3: Are there results about the characteristics of patients who did not fully complete the EQ-5D-5L? Such information can help us understand why patients did not use it.

Answer to comment 3: Our original intention was to collect demographic and clinic information of the patients who declined to answer EQ-5D-5L, to determine if there were any significant differences between responders and non-responders. However, the research ethics board required that non-responders also sign a consent form for chart review and almost all patients who declined to participate in the study also declined to sign that informed consent. Nevertheless, we still collected reasons for declining study participation in 87 out of 169 patients (Table 1). As for the patients who agreed to participate in the study, only 4 did not answer EQ-5D-5L. In the other 14 patients, for whom data was lost because of connectivity issues, EQ-5D-5L answers were complete. All patients who answered the questionnaire, answered all questions.

# Comment 4: I see that the authors have published a similar paper, what are the similarities and differences? https://www.annalsofoncology.org/article/S0923-7534(20)38390-3/fulltext

Answer to comment 4: The link found by the reviewer is an abstract of a poster presentation at the European Society for Medical Oncology Congress. We are now submitting the manuscript for publication for the first time (no similar papers concerning this work have been published before).

# Comment 5: Do the authors think that the use of tablet computers affected the use of the EQ-5D-5L? 

Would it be possible that willingness to complete the instrument may be affected not only by the EQ-5D-5L but also by computer use? Those patients who did not complete it because of the computer?

Answer to comment 5: We address this question in Table 1 and in the Discussion section. Only 6 women (7%) cited the use of tablets as a reason for declining study participation. Patients in cancer centres in Ontario were asked to answer symptom screening and functional status questionnaires in electronic form (computer, tablets, cell phones) prior to the implementation of our study. Accordingly, we did not expect that the use of tablets would represent a novel barrier for them.

# Comment 6: The discussion should be significantly improved. Implications for practice and policy can be included.

Answer to comment 6: The discussion was reviewed accordingly to the comments of both reviewers (please see also points below).

# Comment 7: What does this mean “We excluded patients based on literacy.”?

Answer to comment 7: Sentence was changed to “We excluded patients who were not English literate”.

# Comment 8: A conclusion section is missing. Please provide a conclusion section.

Answer to comment 8: A conclusion section was provided.

# Comment 9: I suggest to use a passive voice for the first sentence of the paper, for example, “A cross-sectional study was conducted to investigate the feasibility …”

Answer to comment 9: The sentence was changed as suggested.

Reviewer 2:

# Comment 1: If feasibility was the sole objective of this study then one might be forgiven for wondering about the need for a research project; completion rate and/or willingness to complete alone in an observational setting would yield an answer. But this manuscript goes well beyond feasibility and provides descriptive material that speaks to the performance of EQ-5D-5L in ambulatory patients in a routine hospital clinic setting. It is way more that “collecting utilities”. The title could really better reflect this significance.

Answer to comment 1: The title of the manuscript was changed to “Implementing routine collection of EQ-5D-5L in a breast cancer outpatient clinic”.

# Comment 2: The rationale for collecting utilities is restricted exclusively to their use in economic evaluation. QALYs are not (and possibly never will be) used to make operational decisions in a clinical setting about individual patients.

Answer to comment 2: The Introduction and Methods sections of the manuscript were reviewed to make it clearer that the purpose of the study was not to use the results of EQ-5D to make decisions about individual patients, but to obtain data to inform drug and health technology reimbursement decision making. We have added the following text to the introduction: “In our context, the purpose of EQ-5D collection was to obtain data to inform drug and health technology reimbursement decision making. EQ-5D scores, although available to clinicians, were not intended to be use for clinical decision making.”

# Comment 3: P4-Line79 refers to EQ-5D as “the most widely used preference-based HRQOL measure”. This is incorrect. EQ-5D is fundamentally a generic measure of health status (aka HrQoL) defined by a classification system that describes health states. These states can be scored/valued/weighted in several different ways – ONE of which is to apply social preference weights. In that highly restricted form alone (ie suitable for QALY computation) it is indeed preference-based. However, EQ-VAS which sits alongside the self-reported EQ-5D classifier, represents a wholly different perspective (the patient themselves) and expresses a value for their health on a 0-100 metric that has absolutely no relationship to the weighted index form. Zero on the EQ-VAS indicates worst possible health status – not “dead”.

Answer to comment 3: We removed the sentence “the most widely used preference-based HRQOL measure”. The EQ-5D-5L questionnaire is described in the methods section only.

# Comment 4: To be honest, the Lines 66 -72 could easily be omitted without loss. Some of the sentiments might be accommodated within the discussion section.

Answer to comment 4: Lines 66-72 of the manuscript (lines 77-83 of the revised manuscript) explain the need for collecting utilities at the population level and we feel that those lines are essential to the rationale of the study. Therefore, we chose not to omit that paragraph.

# Comment 5: The final paragraph on page 4 is better placed in the Methods section on the next page. 

Answer to comment 5: We moved the paragraph to the Methods section.

# Comment 6: In re-reading the introduction the emphasis on economic applications skews the potential usefulness of this paper. Start with the patient and ask yourself which is more relevant to YOU – your self-reported health status on the 5 dimensions and your EQ-VAS, or the social preference-weighted index based on the general population’s views about YOUR health.

Answer to comment 6: While the 5 dimensions of the EQ-5D and the EQ-VAS might be relevant to individual patient care, and the results are worthy of being presented and discussed, the main purpose of the study was to determine if patients would accept to answer EQ-5D, even if the results would not be used to benefit them personally. We have included the following sentence (line 99-101 of the revised manuscript): “Thus, we were interested in assessing feasibility in the context when data collection was not anticipated to have direct individual benefit for respondents.”. The validity of EQ-5D in breast cancer patients will be the subject of a subsequent manuscript we are preparing for publication.

# Comment 7: Line 96 “Symptom Screening and Functional Status questionnaires in waiting room kiosks” These instruments may have worldwide recognition but I was unable to find any reference in the citations or on the web – a brief description would help. The reason? If we are using completion rates then we need to know more about the context and presentation – are waiting room kiosks a standard feature of Canadian hospitals as they are unknown to this reviewer. How long on average does it take to complete these standard questionnaires? What was the order of presentation – was it randomised? Discontinuation of EQ-5D through fatigue/boredom after a 10-minute slog through preceding questionnaires would be a different matter than rejecting it when it was offered first.

Only 61% of patients completing EQ-5D also completed the screening questionnaires (Line 194) so it would be helpful to know more about the precise circumstances under which EQ-5D was completed. How was the timing of EQ-5D completion carried out?

Answer to comment 7: A brief description of the Symptom Screening questionnaire (ESAS) and the Functional Status question are included in lines 122 to 130 of the revised manuscript. Waiting room kiosks exist in all regional cancer centres in Ontario (lines 116 and 117 of the revised manuscript). The study procedures, including the order of presentation of the questionnaires (first the standard of care questionnaires and then EQ-5D) is explained in Figure 1, which was also added to the manuscript.

# Comment 8: Line 120 questionnaire at each clinic visit. Does this mean at subsequent clinic visits? In which case this could be a good proxy for “satisfaction” with EQ-5D (or acceptance of it) – a positive performance attribute.

Answer to comment 8: The expression “each clinic visit” was changed to “subsequent clinic visits” for clarity.

# Comment 9: The classification into health “states” according to disease severity at recruitment is a complication. EQ-5D defines health states, so why not use the term disease severity for the patient’s baseline clinically-assessed health status? In point of fact isn’t this classification better thought of as an index of chronicity – time since initial diagnosis?

Answer to comment 9: We agree with the reviewer and have replaced “breast cancer health states” with “breast cancer states”. Those five breast cancer states were defined a priori and were considered both relevant to clinical practice and economic modeling. Our goal was to find utilities for those breast cancer states. We feel that neither disease severity nor chronicity are adequate descriptors of these states, since patients in states 1 to 4 all have or had curable disease – they differ only in time since the initial diagnosis – whereas patients in state 5, with metastatic disease, can also present with varying levels of disease severity and timing.

# Comment 10: Was all clinically relevant information on patients tested for its association with completion/satisfaction - might the patient's "journey" / treatment pathway be associated with EQ-5D performance? Supplementary free text includes interesting clues suggestive of further investigation - if not by the authors then by others. It would be really good to promote some of this verbatim material to the main text and to have some of its implications explored in the discussion. Surely this is mainstream to the examination of feasibility of routine HrQoL more generally - hence its wider value here.

Answer to comment 10: The most relevant supplementary tables were moved to the results section in the main text and discussion was updated to explore those results.

# Comment 11: Line 151 How was the 70% threshold for completion feasibility established, rule of thumb or external reference? The references cited in the discussion maybe?

Answer to comment 11: The reference for the 70% threshold was added to the sample size calculation section of the manuscript (lines 197 to 199 of the revised manuscript). The 70% threshold was based in the provincial standard for performance in symptom screening for each Regional Cancer Centre in Ontario.

# Comment 12: Analysis with such rich patient data requires careful management. The seeming rush to report utilities should not force out the fundamental material that has been relegated to supplementary tables. Table S2 indicates 2 issues – firstly the classification of patients by disease severity is not helpful as it stands with n=5 for HS2; secondly for the evident high rates of no problem responses for 3/5 dimensions in the survey sample as a whole. The reporting of EQ-5D in its index form effectively conceals this facet of its performance.

Answer to comment 12: Table S2 was moved to the main text (Table 3 of the revised manuscript) and the issues related to patient classification and the high rates of no problem responses discussed in the discussion section (lines 371-372 and 394-397).

# Comment 13: If patient completion rates and degree of acceptance/satisfaction are the primary indicators in this study, then why not exploit the EQ-5D classifier (dichotomize problems none/any) and test whether self-reported problems (at any level) are likely to reduce “performance”? If such a non-random influence were apparent then there are clear implications !!

Answer to comment 13: We added the variables “Problems” / “No problems” in the EQ-5D classifier as predictive variables to the willingness of continuing to answer EQ-5D at subsequent clinic visits in an exploratory analysis presented in the supporting information file (we include this as an exploratory analysis because it was not prespecified).

# Comment 14: If both index and EQ-VAS are to be reported, then it seem

---

## [Decision Letter · Decision Letter 1]

7 Nov 2023

PONE-D-23-18133R1Feasibility of routine collection of health state utilities using EQ-5D-5L in a breast cancer outpatient clinicPLOS ONE

Dear Dr. Torres,

Thank you for submitting your manuscript to PLOS ONE. After careful consideration, we feel that it has merit but does not fully meet PLOS ONE’s publication criteria as it currently stands. Therefore, we invite you to submit a revised version of the manuscript that addresses the points raised during the review process.

We look forward to receiving your revised manuscript.

Kind regards,

Elisa Ambrosi

Academic Editor

PLOS ONE

Reviewers' comments:

Reviewer's Responses to Questions

**Comments to the Author**

1. If the authors have adequately addressed your comments raised in a previous round of review and you feel that this manuscript is now acceptable for publication, you may indicate that here to bypass the “Comments to the Author” section, enter your conflict of interest statement in the “Confidential to Editor” section, and submit your "Accept" recommendation.

Reviewer #1: (No Response)

Reviewer #2: All comments have been addressed

2. Is the manuscript technically sound, and do the data support the conclusions?

Reviewer #1: Yes

Reviewer #2: Yes

3. Has the statistical analysis been performed appropriately and rigorously? 

Reviewer #1: Yes

Reviewer #2: Yes

4. Have the authors made all data underlying the findings in their manuscript fully available?

Reviewer #1: Yes

Reviewer #2: Yes

5. Is the manuscript presented in an intelligible fashion and written in standard English?

Reviewer #1: Yes

Reviewer #2: Yes

6. Review Comments to the Author

Reviewer #1: I've observed that the authors have supplied some information for the my previous comment (see below) in the submission system, but it would be great to include the review contents in the manuscript, either in the literature review section or the discussion section. Please revise the manuscript to include them.

# Comment 2: There are similar studies, please review them and describe the

similiarties and differences between the existing studies and the present study:

https://hqlo.biomedcentral.com/articles/10.1186/s12955-022-02047-0

https://www.thieme-connect.com/products/ejournals/html/10.1055/s-0043-116223

https://www.sciencedirect.com/science/article/pii/S2212109918300487

Reviewer #2: Thank you for your constructive responses.

This box requires a MINIMUM character count .... so please excuse this additional text which is mere padding to fulfil this absurd requrement

7. PLOS authors have the option to publish the peer review history of their article (what does this mean?). If published, this will include your full peer review and any attached files.

Reviewer #1: No

Reviewer #2: **Yes: **Paul Kind

---

## [Author Response · Author response to Decision Letter 1]

22 Dec 2023

Sofia Torres

sofia.cidtorres@mail.utoronto.ca

17 November 2023

Elisa Ambrosi, PhD, Academic Editor

Plos One

Dear Dr. Ambrosi and reviewers,

On behalf of my colleagues, as corresponding author, I would like to thank the editor and external reviewers for the thoughtful review of our manuscript entitled “Implementing routine collection of EQ-5D-5L in a breast cancer outpatient clinic”.

The paper was reviewed, and the only additional comment by reviewer 1 addressed. Reviewer 2 had no further comments. All the following alterations are tracked and explained on the reviewed manuscript:

Reviewer 1: Comment: “I've observed that the authors have supplied some information for the my previous comment (see below) in the submission system, but it would be great to include the review contents in the manuscript, either in the literature review section or the discussion section. Please revise the manuscript to include them.

There are similar studies, please review them and describe the similiarties and differences between the existing studies and the present study: https://hqlo.biomedcentral.com/articles/10.1186/s12955-022-02047-0
https://www.thieme-connect.com/products/ejournals/html/10.1055/s-0043-116223
https://www.sciencedirect.com/science/article/pii/S2212109918300487”

Answer to comment: As explained in our previous answer, our discussion already included several studies we considered relevant for the discussion (lines 375 to 386). The studies mentioned by the reviewer have significant differences in terms of objectives, methodology, population included and setting, compared with our study. Those differences make them sufficiently dissimilar that we do not

feel that we need to include them either in the literature review or the discussion. Nevertheless, we added a paragraph acknowledging those differences in the Introduction section (lines 85 to 90 of the revised manuscript).

I hope to have answered all the concerns raised by the reviewers. Please feel free to contact me if you have any questions or require additional information. My fellow authors and I thank you for considering our manuscript for publication in Plos One.

Kind regards,

Sofia Torres

---

## [Decision Letter · Decision Letter 2]

22 Jan 2024

PONE-D-23-18133R2Implementing routine collection of EQ-5D-5L in a breast cancer outpatient clinicPLOS ONE

Dear Dr. Torres,

Thank you for submitting your manuscript to PLOS ONE. After careful consideration, we feel that it has merit but does not fully meet PLOS ONE’s publication criteria as it currently stands. Therefore, we invite you to submit a revised version of the manuscript that addresses the points raised during the review process.

We look forward to receiving your revised manuscript.

Kind regards,

Elisa Ambrosi

Academic Editor

PLOS ONE

Journal Requirements:

Reviewers' comments:

Reviewer's Responses to Questions

**Comments to the Author**

1. If the authors have adequately addressed your comments raised in a previous round of review and you feel that this manuscript is now acceptable for publication, you may indicate that here to bypass the “Comments to the Author” section, enter your conflict of interest statement in the “Confidential to Editor” section, and submit your "Accept" recommendation.

Reviewer #1: All comments have been addressed

Reviewer #2: (No Response)

2. Is the manuscript technically sound, and do the data support the conclusions?

Reviewer #1: Yes

Reviewer #2: Partly

3. Has the statistical analysis been performed appropriately and rigorously? 

Reviewer #1: Yes

Reviewer #2: Yes

4. Have the authors made all data underlying the findings in their manuscript fully available?

Reviewer #1: Yes

Reviewer #2: Yes

5. Is the manuscript presented in an intelligible fashion and written in standard English?

Reviewer #1: Yes

Reviewer #2: Yes

6. Review Comments to the Author

Reviewer #1: The authors have addressed all comments. The quality of the paper has been improved. I do not have further comments.

Reviewer #2: Thank you for clarifying the focus of the data collection - namely to inform HTA decision-making.

At one level your study demonstrates a degree of feasibility in administering EQ-5D in a clinic setting.

However, around 30% of eligible patients declined to participate (hence likely skewed results, favouring the better educated, less sick patients).

Additionally, as noted in the discussion, of those who did participate, there was a "surprisingly" high rate of patients who reported no problem on ALL dimensions (ie by definition -being in full health).

SO ... whilst demonstrating practical FEASIBILITY the study also highlights the issue of what might be called "legitimacy". In essence. does EQ-5D "work" in this clinic setting and/or for this condition. Concerns around these issues are only briefly alluded to in the Discussion and one might be forgiven for not confronting them here.

HOWEVER, the Conclusion (and Abstract) claim that the study indicates that it is

feasible, ..... to obtain up-to-date population-based HRQOL data that could be used to

inform drug and health technology reimbursement and decision making in Ontario.

Frankly, this statement is at odds with the evidence presented in this study. It is sufficiently wide of the mark that even at this late stage I would urge the authors to consider addressing it. In all other respects the manuscript is fine - of course there can be differences based on style, ethos, beliefs etc .... but this conclusion risks being cited as demonstrating MORE THAN PRACTICAL FEASIBILITY. Given the uncertainty regarding the performance of EQ-5D and patient refusal rates, it would be the height of folly to unconditionally consider this as the basis for obtaining (patient) HrQoL data to inform high-level decision-making.

7. PLOS authors have the option to publish the peer review history of their article (what does this mean?). If published, this will include your full peer review and any attached files.

Reviewer #1: No

Reviewer #2: **Yes: **Paul Kind

---

## [Author Response · Author response to Decision Letter 2]

17 Mar 2024

Sofia Torres 

sofia.cidtorres@mail.utoronto.ca

6 February 2024

Elisa Ambrosi, PhD, Academic Editor 

Plos One 

Dear Dr. Ambrosi and reviewers, 

On behalf of my colleagues, as corresponding author, I would like to thank the editor and external reviewers for the thoughtful review of our manuscript entitled “Implementing routine collection of EQ-5D-5L in a breast cancer outpatient clinic”.

The paper was reviewed. Reviewer 1 had no further comments and considered that all comments were addressed. Reviewer 2 had additional comments which we have addressed in this review. All the following alterations are tracked and explained on the reviewed manuscript:

Reviewer 2:

Comment: “Thank you for clarifying the focus of the data collection - namely to inform HTA decision-making. At one level your study demonstrates a degree of feasibility in administering EQ-5D in a clinic setting. However, around 30% of eligible patients declined to participate (hence likely skewed results, favouring the better educated, less sick patients). Additionally, as noted in the discussion, of those who did participate, there was a "surprisingly" high rate of patients who reported no problem on ALL dimensions (ie by definition -being in full health). SO ... whilst demonstrating practical FEASIBILITY the study also highlights the issue of what might be called "legitimacy". In essence. does EQ-5D "work" in this clinic setting and/or for this condition. Concerns around these issues are only briefly alluded to in the Discussion and one might be forgiven for not confronting them here. HOWEVER, the Conclusion (and Abstract) claim that the study indicates that it is feasible, ..... to obtain up-to-date population-based HRQOL data that could be used to inform drug and health technology reimbursement and decision making in Ontario. Frankly, this statement is at odds with the evidence presented in this study. It is sufficiently wide of the mark that even at this late stage I would urge the authors to consider addressing it. In all other respects the manuscript is fine - of course there can be differences based on style, ethos, beliefs etc .... but this conclusion risks being cited as demonstrating MORE THAN PRACTICAL FEASIBILITY. Given the uncertainty regarding the performance of EQ-5D and patient refusal rates, it would be the height of folly to unconditionally consider this as the basis for obtaining (patient) HrQoL data to inform high-level decision-making.”

Answer to comment: Thank you for the additional comments. We removed the following sentences respectively from the abstract and the conclusions: “in order to obtain up-to-date population-based HRQOL data that could be used to inform drug and health technology reimbursement and decision making in Ontario.” (lines 53-54) and, “in order to obtain up-to-date population-based HRQOL data that could be used to inform drug and health technology reimbursement and decision making in Ontario.” (lines 448-450).

We made the following changes in the Abstract (lines 52-56): “Tablet-based collection of EQ-5D-5L in the context of routine clinical practice proved to be feasible. However, many patients declined study participation or reported being in full health, raising concerns about whether this method of collecting EQ-5D adequately represents the health status of all breast cancer patients.”

In the Manuscript we added the following sentence to the Discussion (lines 387-388): “or whether our study population was representative of all breast cancer patients.”. The Conclusions were also updated (lines 446-452): “In conclusion, our study demonstrated that tablet-based collection of EQ-5D-5L in the context of routine clinical practice was feasible for breast cancer patients. EQ-5D-5L can be easily administered, in addition to the Symptom Screening and Functional Status questionnaires. However, many patients declined study participation or reported being in full health, raising concerns about whether this method of collecting EQ-5D adequately represents the health status of all breast cancer patients.”

I hope to have answered all the concerns raised by the reviewers. Please feel free to contact me if you have any questions or require additional information. My fellow authors and I thank you for considering our manuscript for publication in Plos One. 

Kind regards,

Sofia Torres

---

## [Decision Letter · Decision Letter 3]

2 Jul 2024

Implementing routine collection of EQ-5D-5L in a breast cancer outpatient clinic

PONE-D-23-18133R3

Dear Dr. Torres,

We’re pleased to inform you that your manuscript has been judged scientifically suitable for publication and will be formally accepted for publication once it meets all outstanding technical requirements.

Kind regards,

Ramune Jacobsen

Academic Editor

PLOS ONE

Reviewers' comments:

Reviewer's Responses to Questions

**Comments to the Author**

1. If the authors have adequately addressed your comments raised in a previous round of review and you feel that this manuscript is now acceptable for publication, you may indicate that here to bypass the “Comments to the Author” section, enter your conflict of interest statement in the “Confidential to Editor” section, and submit your "Accept" recommendation.

Reviewer #3: All comments have been addressed

2. Is the manuscript technically sound, and do the data support the conclusions?

Reviewer #3: Yes

3. Has the statistical analysis been performed appropriately and rigorously? 

Reviewer #3: Yes

4. Have the authors made all data underlying the findings in their manuscript fully available?

Reviewer #3: Yes

5. Is the manuscript presented in an intelligible fashion and written in standard English?

Reviewer #3: Yes

6. Review Comments to the Author

Reviewer #3: (No Response)

7. PLOS authors have the option to publish the peer review history of their article (what does this mean?). If published, this will include your full peer review and any attached files.

Reviewer #3: No

---

## [Editor Report · Acceptance letter]

12 Jul 2024

PONE-D-23-18133R3 

PLOS ONE

Dear Dr. Torres, 

I'm pleased to inform you that your manuscript has been deemed suitable for publication in PLOS ONE. Congratulations! Your manuscript is now being handed over to our production team.

Kind regards, 

on behalf of

Dr. Ramune Jacobsen 

Academic Editor

PLOS ONE